# Evolution of Microstructure and Mechanical Properties of LM25–HEA Composite Processed through Stir Casting with a Bottom Pouring System

**DOI:** 10.3390/ma15010230

**Published:** 2021-12-29

**Authors:** Mekala Chinababu, Nandivelegu Naga Krishna, Katakam Sivaprasad, Konda Gokuldoss Prashanth, Eluri Bhaskara Rao

**Affiliations:** 1Advanced Materials Processing Laboratory, Department of Metallurgical and Materials Engineering, National Institute of Technology, Tiruchirappalli 620015, India; mecchina.413@gmail.com; 2Department of Mechanical Engineering, Vishnu Institute of Technology, Bhimavaram 534202, India; nagakrishna.n@vishnu.edu.in; 3Department of Mechanical and Industrial Engineering, Tallinn University of Technology, 19086 Tallinn, Estonia; 4Erich Schmid Institute of Materials Science, Austrian Academy of Science, 8700 Leoben, Austria; 5Centre for Biomaterials, Cellular, and Molecular Theranostics, Vellore Institute of Technology, Vellore 632014, India; 6Naval Science and Technological Laboratory, Visakhapatnam 530027, India; eluribhaskarrao@gmail.com

**Keywords:** metal matrix composite, LM25 alloy, stir casting, high entropy alloy, mechanical properties

## Abstract

Aluminum matrix composites reinforced by CoCrFeMnNi high entropy alloy (HEA) particulates were fabricated using the stir casting process. The as-cast specimens were investigated by X-ray diffraction (XRD), scanning electron microscopy (SEM), energy dispersive spectroscopy (EDS), and transmission electron microscopy (TEM). The results indicated that flake-like silicon particles and HEA particles were distributed uniformly in the aluminum matrix. TEM micrographs revealed the presence of both the matrix and reinforcement phases, and no intermetallic phases were formed at the interface of the matrix and reinforcement phases. The mechanical properties of hardness and tensile strength increased with an increase in the HEA content. The Al 6063–5 wt.% HEA composite had a ultimate tensile strength (UTS) of approximately 197 MPa with a reasonable ductility (around 4.05%). The LM25–5 wt.% HEA composite had a UTS of approximately 195 Mpa. However, the percent elongation decreased to roughly 3.80%. When the reinforcement content increased to 10 wt.% in the LM25 composite, the UTS reached 210 MPpa, and the elongation was confined to roughly 3.40%. The fracture morphology changed from dimple structures to cleavage planes on the fracture surface with HEA weight percentage enhancement. The LM25 alloy reinforced with HEA particles showed enhanced mechanical strength without a significant loss of ductility; this composite may find application in marine and ship building industries.

## 1. Introduction

Metal matrix composites (MMCs) are replacing monolithic alloys in various structural and other applications, due to their enhanced properties, such as high specific strength, mechanical and tribological properties. Aluminum (Al) based MMCs are widely used for structural applications in the automotive, aerospace, and chemical industries. This is attributed to their high strength-to-weight ratio, high elastic modulus, wear resistance, and corrosion resistance [1,2]. To enhance the mechanical properties of Al-based composites, they are reinforced with Al_2_O_3_, B_4_C, SiC, TiB_2_, or TiC. However, there are major drawbacks of these ceramic-reinforced Al-based composites, including supporting agglomerations, particle fragmentation, porosity, and cracking [3]. In addition, the vast variation of the thermal expansion coefficient between the ceramic particles and metal matrix, inferior wettability at the interface, and the reaction at the interface reduce the properties of MMCs [4,5].

To overcome the above drawbacks and improve the structural properties of Al-based composites, a new reinforcement system with metallic elements was investigated [6]. Instead of a single metal, a multi-component system was preferred as reinforcement due to its superior mechanical, thermal, and corrosion properties. Yeh et al. [7] termed these multi-component systems as high entropy alloys (HEAs). HEAs are a new class of materials designed as an alloy comprised of five or more principal elements, with concentrations ranging from 5% to 35% [8]. In HEAs, mixing entropies enables the formation of random solid-solution phases instead of intermetallic phases. Therefore, HEAs exhibit superior plasticity. In addition, the many elements of different atomic radii contained in HEAs result in severe lattice distortion and helps them to possess high strength [9]. There are various reported beneficial properties of the HEA system, such as increased strength and hardness, good corrosion and oxidation resistance, wear and fatigue with good thermal stability, magnetic properties, and increased stability at elevated temperatures. In addition, the four main effects of HEAs, i.e., high entropy, sluggish diffusion, severe lattice distortion, and cocktail effects, affect phase transformation, microstructure, and mechanical properties more significantly than low entropy alloys [10]. When compared with conventional ceramic reinforcements, the CoCrFeMnNi HEA particles used as reinforcement in the present work were expected to exhibit better mechanical strength without losing ductility.

Preliminary work has been carried out considering HEAs as a reinforcement phase in MMCs. Wang et al. [11] used FeNiCrCoAl_3_ particles in 2024 Al matrix composites and obtained a compressive strength of 710 MPa. Liu et al. [12] reported the influence of a transition layer on AlCoCrFeNi HEA particle-reinforced Al matrix composites. Using transmission electron microscopy, the authors observed that the interface layer had an face centered cubic (*fcc*) structure. In another study, AlCoCrCuFeNi HEA as reinforcement in an AZ91D matrix composite coating was prepared through laser surface forming [13]. It was observed that HEA reacted with the magnesium alloy matrix and formed a new phase, which resulted in a significant improvement in the wear resistance of the coating. Chen et al. [14] considered the powder metallurgy route to fabricate a copper matrix composite with AlCoNiCrFe as the reinforcement phase and found that the yield strength of composites was enhanced by approximately 160% compared with the yield of the Cu matrix, and the elongation increased by approximately 15%. Ananiadis et al. [15] studied the microstructure and corrosion performance of Al matrix composites reinforced with refractory (MoTaNbVW) HEA particulates. The composite was fabricated through the powder metallurgy route, and it was reported that increasing the volume of the reinforcing phase enhanced the composite’s hardness. Similarly, there are numerous reports published on composites where HEA is used [16,17,18,19,20]. The majority of these reports show powder metallurgy as the preferred production route, since reinforcing the particles may be easier when using powder metallurgy than when using other casting options [21,22,23]. In addition, some of these reports also involve additive manufacturing as next generation manufacturing processes to fabricate such composites [24,25]. However, casting is one of the most cost-effective methods compared with the novel additive manufacturing routes. Hence, the present manuscript used the cost-effective method of casting to process and fabricate HEA-reinforced metal matrix composites. Moreover, HEA particles are said to show sluggish diffusion; hence, they will not readily absorb moisture nor oxidize easily. Thus, this will help in avoiding any processing of the reinforcement powders before their introduction into the melt. In addition, since the HEA particles have structural stability, they may not react with the matrix when introduced as reinforcement. Since all the elements in the HEA are metallic in nature, they will offer better wettability than the ceramic reinforcement. Considering all these advantages, CoCrFeMnNi HEA particles were used as reinforcement in the present work, and LM25 Al alloy was taken as the matrix. LM25 is a cast alloy of aluminum that has superior mechanical strength and improved corrosion resistance. LM25 Al alloy is generally used in packaging, chemical, marine, and mobility engineering, where it is used for wheels, cylinder blocks and heads, and other engine and body castings. The current work focuses on the fabrication of Al-MMCs by a stir casting process with a bottom pouring system. The various properties of the resultant composites were studied in detail.

## 2. Materials and Methods

The metal matrix composites were fabricated and synthesized through a bottom pouring stir casting unit. Crystalline CoCrFeMnNi HEA particles were used as the reinforcement phase and commercially pure Al and LM25 alloy were used as the matrix material. The matrix material (AL/LM25) was melted in a mechanized induction type stir casting furnace (Figure 1) with a tapered bottom and a bottom pouring facility (Indfurr, Chennai, India) at 800 °C. In order to achieve the desired dispersion in the matrix, the HEA reinforcement particles were then introduced into the melt, and the melt was stirred at 400 rpm for 5–10 min. The parameters were chosen partially based on the literature and also on prior casting experience, especially in the fabrication of composites. Once optimum stirring was realized and the reinforcement particles were dispersed uniformly in the melt, the nozzle at the bottom of the furnace was then opened to allow the melt to fill the rectangular die of 120 mm × 150 mm × 20 mm placed at the bottom of the furnace. Sufficient time was allowed for the composite material to solidify and cool down before sampling for metallographic preparation.

Representative composite pieces were cut from the cast bar and subjected to standard metallographic sample preparation techniques (using standard metallographic procedures), and finally were etched with Kellar’s reagent. The microstructures of the composites were observed under an Olympus GX41 inverted microscope. A field-emission scanning electron microscope (FESEM, Gemini-300, ZEISS, Jena, Germany) with a combined energy-dispersive X-ray spectroscope (EDS) feature identified elemental composition as well as surface morphology. The X-ray diffraction analysis was carried out using a Rigaku Ultima IV XRD unit (Rigaku, Stuttgart, Germany) (with Cu-Kα radiation: λ: 1.5406 Å), which functioned at 30 mA and 40 kV by recording diffraction patterns at a scan rate of 0.01° from 2θ ranging between 20° to 100° to confirm the crystal structure and phases produced. In addition, the specimens underwent a microhardness test for a load of 0.2 kg and a dwell time of 5 sec using a Shimadzu microhardness tester HMV-G20, and the values indicated are an average of the 6 to 8 values considered for the specimen.

Tensile tests were carried on the micro tensile dog bone type specimen with a strain rate of 5 × 10^−4^ s^−1^ according to the ASTM-E08-16 standard using a Tinius Olsen tabletop tensile testing machine (model H25KL, TINIUS OLSEN, Redhill, UK). The fractography of the fractured tensile specimens was evaluated with scanning electron microscopy (SEM) using a TESCAN microscope (Oxford). Specimens were prepared for transmission electron microscopy (TEM) as 0.1 mm foil with a repeated series of disk and emery polishing. From that, a 3 mm piece was cut with a disk punch. Then, the 3 mm piece was placed in a grinder wheel and its thickness was reduced with a disk grinder. The resulting thin foil was placed in a Gatan precision ion polishing system (PIPS) that comprises double penning ion guns with a beam diameter of 350 lm. The adjustment of the beam diameter was made by argon gas; the ion milling was carried out until good electron transparency was obtained. Vacuum pressure was maintained in the order of 8–9 × 10^−6^ Torr pressure. The TEM images of the specimens were examined through JOEL JEM 2100 equipment (JEOL, Freising, Germany) with an accelerating voltage of 20 kV and very high magnification.

## 3. Results and Discussion

### 3.1. Evolution of Microstructure

Figure 2 shows the distribution of reinforcing particles within the Al alloy matrix for the various composite systems. It can be observed that the actual reinforcement content is in better agreement with the composition, i.e., the higher the composition, the greater the actual reinforcing particle content. Concerning the particle distribution, it can be observed from Figure 2 that the particle distribution is homogeneous for the different composite systems. In the Al 6063–5 wt.% HEA (Figure 2a), the HEA particles were distributed uniformly throughout the Al matrix. In the LM25–5 wt.% HEA (Figure 2b), HEA particles and flake-like silicon particles were distributed nearly uniformly in the Al matrix. In the LM25–10 wt.% HEA (Figure 2c), HEA particles and flake-like silicon particles (silicon being one of the major elements present in LM25 alloy, which was used as the matrix material in the present work) were distributed nearly uniformly in the Al matrix. Still, the silicon particles’ percentage slightly decreased when compared with the LM25–5 wt.% HEA. For all the three composite systems, the various particles distributed in the matrix are indicated with arrows.

Figure 3 shows the scanning electron microscopy images of Al-based alloy composites with varying HEA content. The secondary electron mode and the backscattered electron mode images (EBSD) are shown. Regarding the distribution of reinforcement in the matrix, it was observed that the HEA particles were nearly homogeneously distributed in all three different composite systems. Figure 3a,b correspond to the Al 6063–5 wt.% HEA; in this composite, HEA particles with an irregular polygonal shape were distributed in the Al matrix, identified and indicated in Figure 3. Figure 3c,d correspond to the LM25–5 wt.% HEA; in this composite, the flake-like silicon particles and the reinforcement phase of HEA particles were nearly equally distributed in the Al matrix due to their similar weight percentage (5 to 6%). Figure 3e,f correspond to the LM25–10 wt.% HEA; in this composite, the flake-like silicon particles and the reinforcement phase of HEA particles were nearly uniformly distributed, with the domination of reinforcement HEA particles due to their high weight percentage (10 wt.%) when compared with the Si in the Al matrix. No significant pores were observed. At the same time, the interfacial detachment was not visible. In addition, there was no substantial evidence of a reaction between the matrix and the reinforcement particles at the interfacial areas. This observation is critical because it indicates an absence of potential brittle intermetallic phases at the interface. As reported by Lekatou et al. [26], the presence of intermetallic steps at the interfacial area can cause deterioration in the properties of the composite.

Energy dispersive spectroscopy mapping with line scanning images of Al-based HEA composites with varying weight percentages are shown in Figure 4, Figure 5 and Figure 6. The line scans were measured in the vicinity of a bulk reinforcing particle and the matrix to reveal the elemental distribution. The reinforcing particle consisted of the elements related to the refractory HEA system used in the present work. Al, Co, Cr, Fe, Mn, and Ni elements were homogeneously distributed in the HEA particles and retained their HEA composition even though stir casting through a bottom pouring system took place. No significant interfacial reaction was observed along with the interface of the HEA particles and the matrix. In EDS, the K factor’s magnitude characterizes the element’s content, and its calculation is made according to the ratio standard. If its value indicates less, then the intensity of the component can be eliminated.

Figure 4 shows energy dispersive spectroscopy mapping with line scanning images of the Al 6063–5 wt.% HEA. From the EDS elemental mapping images, we confirmed that the elements present in the given composite consisted of roughly 97% Al matrix, Cr, Fe, and Mn elements with a nearly equal distribution and composition around 1%, and Co and Ni elements, which were negligible in the selected area. Based on the line scanning images, it was observed that the distribution of Co, Cr, Fe, Mn, and Ni in the dispersion phase and the matrix was nearly identical. Figure 5 shows energy dispersive spectroscopy mapping with line scanning images of LM25–5 wt.% HEA. From the EDS elemental mapping images, we confirmed that elements present in the given composite consisted of approximately 95% Al-matrix (roughly 86% Al, 8% Si, and 1% Mg), a nearly equal distribution of Co, Cr, Fe, and Mn elements with a composition of approximately 1%, and a negligible amount of Ni in the selected area. Based on the line scanning images, it was observed that the distribution of Co, Cr, Fe, Mn, and Ni in the dispersion phase and the matrix was nearly identical.

Figure 6 shows EDS mapping with line scanning images of the LM25–10 wt.% HEA. From the EDS elemental mapping images, we confirmed that the elements present in the given composite consisted of approximately 94% Al matrix (roughly 85% Al, 8% Si, and 1% Mg) with a nearly equal distribution of Cr and Fe (around 2%) and Mn and Ni (around 1%) in the selected area. Based on the line scanning image, it was observed that the distribution of Co, Cr, Fe, Mn, and Ni in the dispersion phase and the matrix was nearly identical, with higher contrast when compared to the LM25–5 wt.% HEA.

Figure 7 presents the transmission electron microscopy images of the Al-based HEA composites with varying reinforcement content. As observed in the figure, no interfacial reaction was visible between the CoCrFeMnNi HEA and the matrix. In addition, it is evident from the selected area electron diffraction (SAED) patterns that no intermetallic phases formed at the CoCrFeMnNi HEA and matrix boundaries. The matrix and the reinforcement phases are identified and indicated in Figure 7 (reinforcement HEA with a rectangular shape and matrix with a circular shape). Similar observations were made by Chen et al. [8]. No reaction was observed between the Cu matrix and AlCoNiCrFe HEA reinforcement at the interface, and the HEA particles in the Cu matrix had an average size of 20 nm. Karthik et al. [27] also reported no significant diffusion between the constituent elements of the HEA particles and the Al matrix or vice versa. They mentioned that the CoCrFeNi HEA system was thermally stable and did not undergo significant grain growth. Figure 7a,b correspond to the bright-field image and its corresponding selected area electron diffraction (SAED) pattern, respectively, of the Al 6063–5 wt.% HEA composite. In this composite, it was observed that the HEA particles were distributed nearly uniformly in the Al matrix. There was no evidence for \the interfacial reaction and the formation of intermetallic compounds between the matrix and HEA reinforcement. Figure 7c,d correspond to the bright-field image and its corresponding selected area electron diffraction (SAED) pattern, respectively, of the LM25–5 wt.% HEA composite. In this composite, it was observed that the HEA particles and Si particles were distributed nearly uniformly in the Al matrix. In this case, there was also no evidence for the interfacial reaction and the formation of intermetallic compounds between the matrix and HEA reinforcement. Figure 7e,f correspond to the bright-field image and its corresponding selected area electron diffraction (SAED) pattern, respectively, of the LM25–10 wt.% HEA composite. In this composite, it was observed that the HEA particles and Si particles were distributed nearly uniformly in the Al matrix, with a higher HEA reinforcement content than Si particles; no interfacial reaction or formation of intermetallic compounds occurred. Yang et al. [28] studied the interface morphology between AlCoCrFeNi particles and a 5083 Al matrix. In their study, they reported that the HEA–5083 Al composite exhibited superior interfacial integrity, and neither micropores nor microcracks were formed at the interface. Similar results were observed in the present work, and there was no major diffusion observed.

### 3.2. Microhardness Studies

The hardness variation for the different composites is shown in Figure 8. It can be observed that with the enhancement of the reinforcement contents, the hardness increased. In the case of the Al 6063–5 wt.% HEA, the hardness was increased by nearly 15% compared with the microhardness of Al 6063 (54 HV_0.20_) reported by S. Najafi et al. [29]. In the case of the LM25–5 wt.% HEA, the hardness increased by approximately 41% as compared with the cast LM25 alloy (55 HV_0.20_). In the case of the LM25–10 wt.% HEA composites, an approximately 67% increase in hardness was observed when compared with the cast LM25 alloy. Similar results were reported by Praveen Kumar et al. [30], where AA 2024 reinforced with 15% HEA showed a 62% improvement in hardness compared to monolithic AA 2024 alloy. They reported that this increase in hardness may be influenced by the reinforcement particles, the refined grain size of the Al alloy matrix, interparticle distance, bonding at the matrix–reinforcement interface, constrained dislocation movement, and higher dislocation density [31].

### 3.3. X-ray Diffraction Analysis

The X-ray diffractograms of the Al6063–5 wt.% HEA, LM25–5 wt.% HEA, and LM25–10 wt.% HEA composite samples are represented in Figure 9. It can be observed that in all cases, two different phases can be identified: an FCC phase with intense high peaks corresponding to the Al matrix and a BCC phase of significantly lower peak intensities corresponding to the HEA reinforcing particles. Compared to the LM25–5 wt.% HEA sample, the peak intensities of the BCC phase are higher in the LM25–10 wt.% HEA sample, corroborating the presence of higher amounts of HEA reinforcement particles. As reported elsewhere [32], the crystallite size and microstrain were calculated using the Williamson–Hall (WH) technique of X-ray diffraction line profile analysis (XRDLPA). The dislocation density corresponding to crystallite size and microstrain values was calculated [32], and the corresponding values are depicted in Table 1. Table 1 shows that the crystallite size for the Al 6063–5 wt.% HEA was 36 nm, LM25–5 wt.% HEA was 37 nm, and LM25–10 wt.% HEA was 35 nm. The dislocation density and lattice strain values increased with an increase in the reinforcement content. Similar behavior was reported by Kumar et al. [33] in their study on Al 7075–Al_2_O_3_ MMCs, where the authors concluded that the hardness of the composites was enhanced with increased filler content.

### 3.4. Mechanical Properties

Figure 10a,b show the stress versus strain and the variation of mechanical properties with sample designation, respectively. From the figure, it was observed that the yield strength (YS), ultimate tensile strength (UTS), and percent elongation (% E) were increased with increasing HEA reinforcement. It can be observed from Figure 10a that the Al 6063–5 wt.% HEA composite had a UTS of approximately 197 MPa with a reasonable ductility of roughly 4%. The LM25–5 wt.% HEA composite had a UTS of approximately 195 MPa. However, the elongation decreased to roughly 3.90%. When the reinforcement content increased to 10 wt.% to LM25 alloy, the UTS reached approximately 210 MPa, and the elongation was confined around 3.30%. Similar results were reported in a study by Li et al. [34], in which the mechanical properties of Al-based MMCs with different contents of AlFeNiCrCoTi HEA (4, 5, and 6 wt.%) were investigated. The authors found that intermetallic compound strengthening, solid solution strengthening, and grain refinement enhanced the strength of the Al-based MMCs. When the concentration of HEA was 4 wt.%, the UTS was 142 MPa, and it increased to 170 MPa for 5 wt.% HEA. Figure 10b shows the variation versus sample designation. From Figure 10b, it was observed that YS and UTS increased with increasing HEA reinforcement, and the elongation decreased with increasing HEA reinforcement. The values of the various properties are indicated in Figure 10b, and the corresponding mechanical property values are depicted in Table 2.

In their work, Karthik et al. [27] reported that the high strength and hardness of the HEA reinforcement exhibits better tensile and compressive properties. A similar result was obtained in the present work, where in the case of the Al 6063–5 wt.% HEA composite, the YS and UTS increased by approximately 86% and 51%, respectively, and the percent elongation decreased by 50%, compared with as-cast LM25 alloy. The work of Schuh et al. [35] attained similar results obtained through arc melting and drop-casting processes for fabrication of an equiatomic CoCrFeMnNi HEA. The composite underwent high-pressure torsion, which induced significant grain refinement in the coarse-grained casting, resulting in a grain size of approximately 50 nm. As a result, the strength was enhanced to 1.95 GPa. In the case of the LM25–5 wt.% HEA composite, YS and UTS increased by approximately 74% and 50%, respectively, and the percent elongation was decreased by approximately 51%. For the LM25–10 wt.% HEA composite, YS and UTS increased by approximately 84% and 61%, respectively, and the percent elongation was decreased by around 60%. Similar results were observed in a study by Wang et al. [36], where high quality aluminum CuZrNiAlTiW high entropy alloy (HEA) composites were fabricated by mechanical alloying and spark plasma sintering (SPS). The HEA powders in the as-milled condition conformed to a single body-centered cubic (BCC) solid-solution phase, and the formation of NiAl-rich B_2_ and WAl_12_ phases occurred in the sintered composites. This was attributed to the high concentration gradient of Al between the matrix and the HEA reinforcement.

The spark plasma sintered Al bulk exhibited lower microhardness and strength, compared with the Al–HEA composite. Of the three fabricated Al–metal matrix composites presented here, the LM25–10 wt.% HEA composite exhibited superior properties due to its higher HEA reinforcement content over the Al 6063–5 wt.% HEA and LM25–5 wt.% HEA. Its YS was slightly reduced; however, its UTS values increased by nearly 6% and percent elongation was decreased by approximately 19% in comparison with the Al 6063–5 wt.% HEA. Furthermore, its YS and UTS values increased by nearly 6%, and 7% and percent elongation was decreased by roughly 15% compared to the LM25–5 wt.% HEA composite. Yang et al. [28] reported that the fabrication of 5083 Al matrix composite was reinforced by submerged friction stir processing (SFSP). When compared to a base metal, the submerged friction stir processed HEA–5083Al composites exhibited enhanced yield stress (YS) and ultimate tensile strength (UTS) by 25% and 32%, respectively, and good ductility (18.9%). In the present work, the obtained results were similar with a slight enhancement in the properties.

### 3.5. Fractography

Figure 11 represents the scanning electron microscopy images of the fracture surface. Figure 11a shows the fracture image of Al 6063–5 wt.% HEA composite. The primary fracture mode in the specimens was ductile, with reduced cavitation; in addition, the reduction of dimples in the fracture and slight cleavage planes (indicated by arrows) were also observed. Figure 11b shows the fracture image of LM25–5 wt.% HEA composite; it can be observed that with the addition of HEA, the cleavage plane appeared on the fracture surface along with the presence of dimples. Figure 11c shows the fracture image of LM25–10 wt.% HEA composite; as the HEA was enhanced from 5 wt.% to 10 wt.%, the cleavage plane appeared on the fracture surface and significantly reduced the number of dimples. Similar results were reported in a study by Li et al. [34], in which they found that the fracture surface of pure Al contained many uniform dimples. With the inclusion of HEA particles in the Al matrix, the dimple size was reduced. For the composite containing 6 wt.% HEA, the fractography revealed blocky intermetallic compounds, and cracks were generated at the tip of the intermetallic compound due to high-stress concentration [34].

## 4. Conclusions

Al 6063 and LM25 matrix composites reinforced with CoCrFeMnNi HEA particles were fabricated through stir casting with a bottom pouring system.Optical and scanning electron microscopy images revealed that the reinforcement particles were distributed homogeneously.From XRD phase analysis, two different phases were observed. The peaks with higher intensities were identified as the FCC phase and correspond to Al. The peaks with significantly lower intensities correspond to the reinforced HEA particles and have a BCC structure.Some mechanical properties, such as microhardness, yield strength, and ultimate tensile strength, were increased with increased HEA reinforcement content. However, ductility was decreased with an increase in HEA reinforcement content.As HEA content was increased, the fracture surface revealed a cleavage plane and a significant reduction in the number of dimples, corroborating the mechanical test results.

## Figures and Tables

**Figure 1 materials-15-00230-f001:**
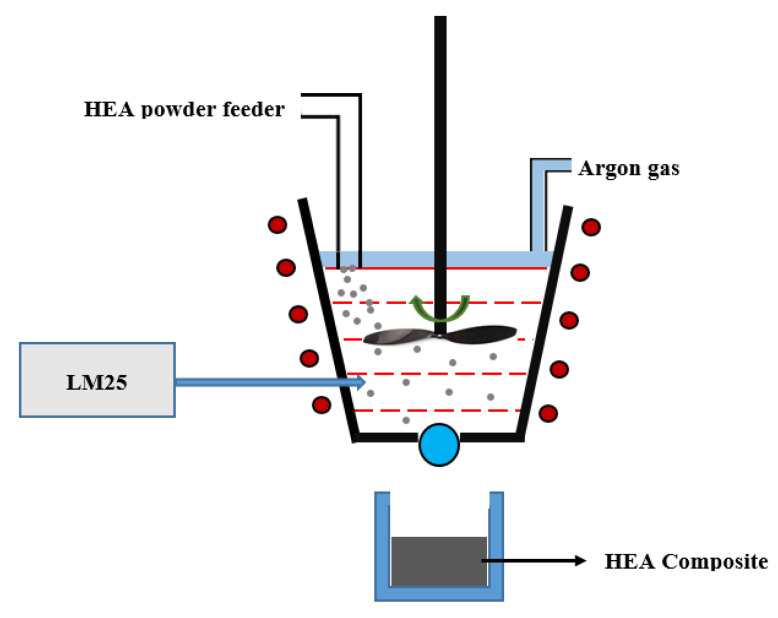
Schematic representation of stir casting furnace and composite production.

**Figure 2 materials-15-00230-f002:**
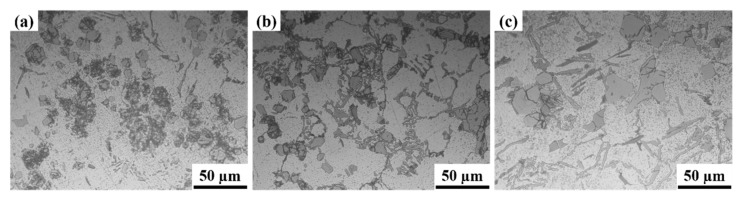
Optical micrographs of the Al-based metal matrix composites with varying concentrations of HEA: (**a**) Al 6063–5 wt.% HEA, (**b**) LM25–5 wt.% HEA, and (**c**) LM25–10 wt.% HEA.

**Figure 3 materials-15-00230-f003:**
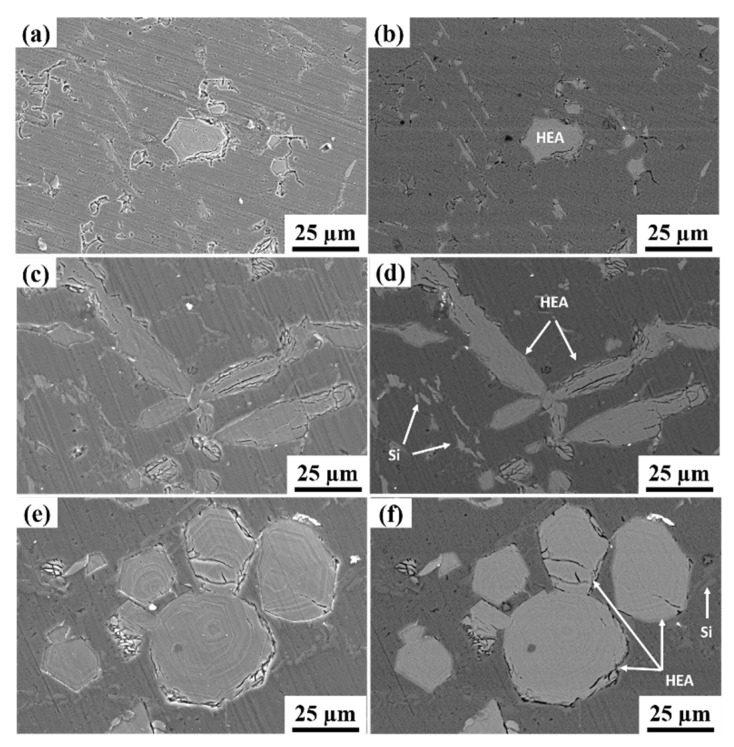
Scanning electron microscopy images of Al 6063–5 wt.% HEA in (**a**) secondary electron mode and (**b**) backscattered electron mode. Scanning electron microscopy images of LM25–5 wt.% HEA in (**c**) secondary electron mode and (**d**) backscattered electron modes. Scanning electron microscopy images of LM25–10 wt.% HEA (**e**) secondary electron mode and (**f**) backscattered electron mode.

**Figure 4 materials-15-00230-f004:**
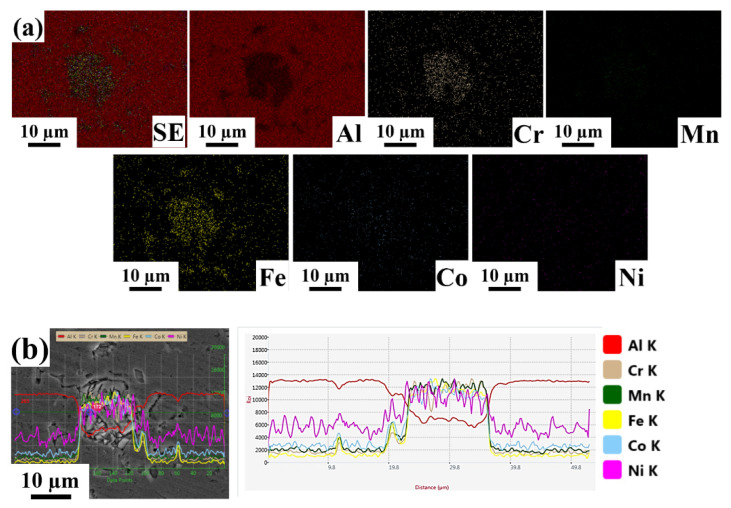
Energy dispersive spectroscopy images of Al 6063–5 wt.% HEA (**a**) Elemental mapping images. (**b**) Line scan image.

**Figure 5 materials-15-00230-f005:**
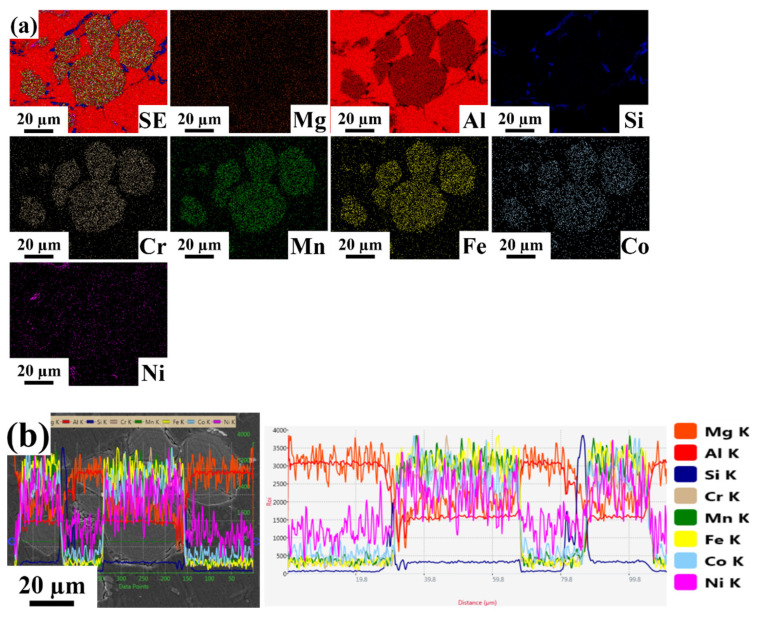
Energy dispersive spectroscopy images of LM25–5 wt.% HEA (**a**) Elemental mapping images. (**b**) Line scan image.

**Figure 6 materials-15-00230-f006:**
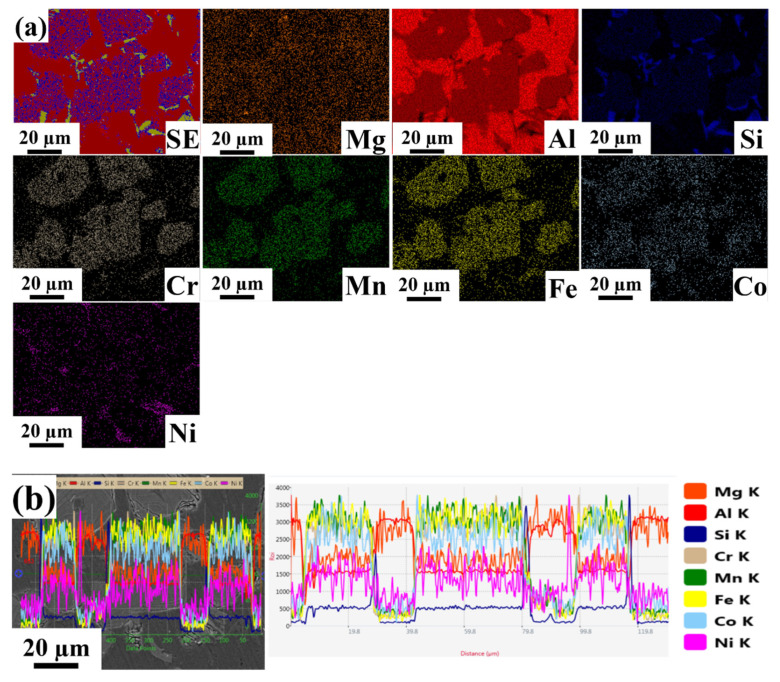
Energy dispersive spectroscopy images of LM25–10 wt.% HEA. (**a**) Elemental mapping images. (**b**) Line scan image.

**Figure 7 materials-15-00230-f007:**
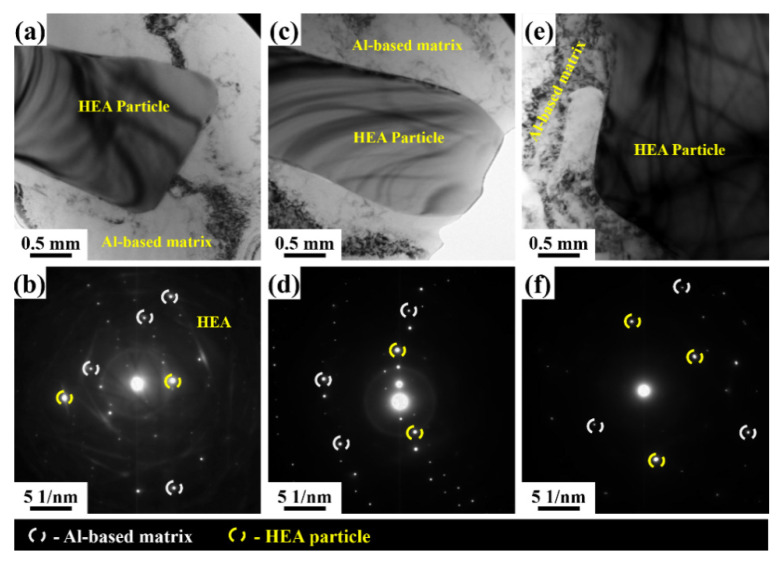
Transmission electron microscopy images of Al 6063–5 wt.% HEA composite: (**a**) bright-field image and (**b**) its corresponding selected area diffraction pattern. Transmission electron microscopy images of LM25–5 wt.% HEA composite: (**c**) bright-field image and (**d**) its corresponding selected area diffraction pattern. Transmission electron microscopy images of LM25–10 wt.% HEA composite: (**e**) bright-field image and (**f**) its corresponding selected area diffraction pattern.

**Figure 8 materials-15-00230-f008:**
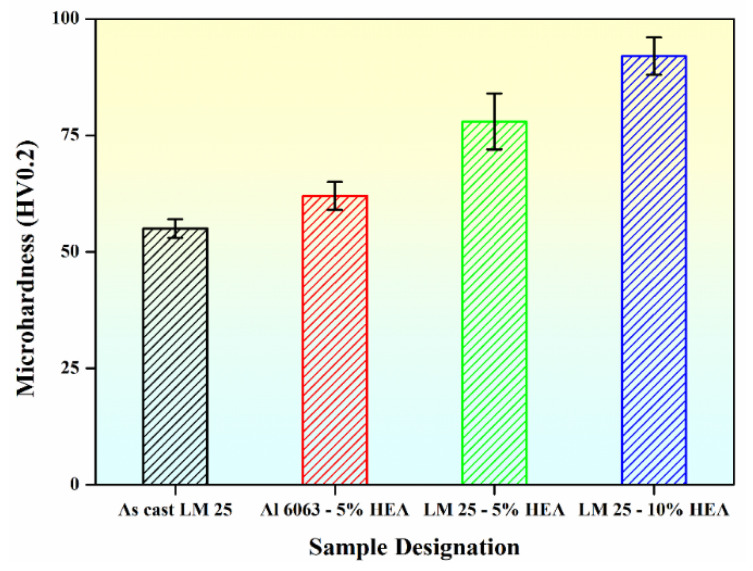
The microhardness values of the Al-based composites as a function of different HEA content.

**Figure 9 materials-15-00230-f009:**
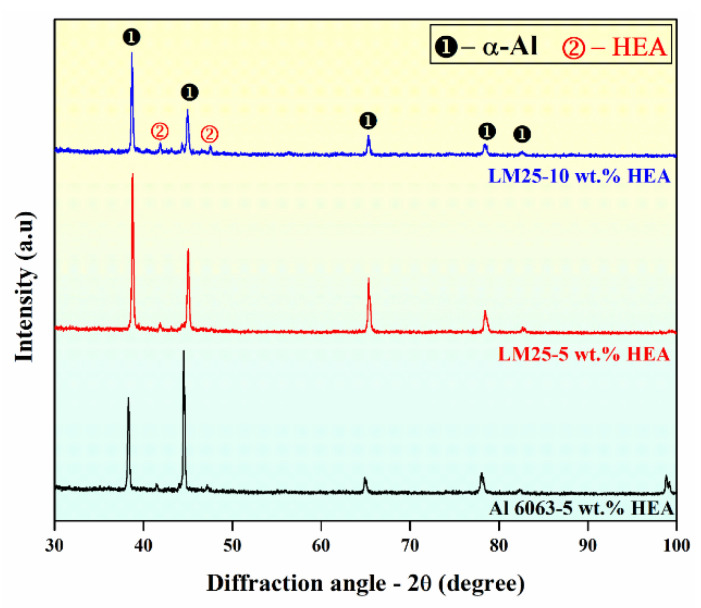
The X-ray diffraction patterns of the Al-based metal matrix composites with various HEA content.

**Figure 10 materials-15-00230-f010:**
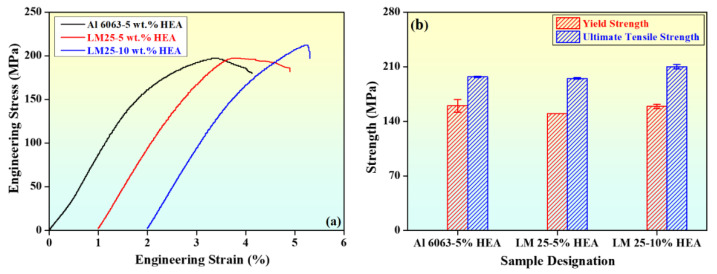
Mechanical properties of the Al-based metal matrix composites with different HEA content. (**a**) Stress vs. strain curve. (**b**) Variation of properties with sample designation.

**Figure 11 materials-15-00230-f011:**
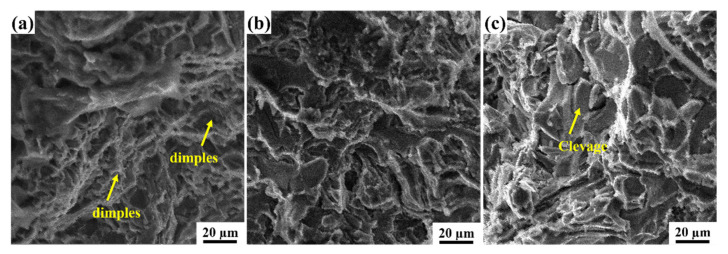
SEM fractographs of composites: (**a**) Al 6063–5 wt.% HEA, (**b**) LM25–5 wt.% HEA, and (**c**) LM25–10 wt.% HEA.

**Table 1 materials-15-00230-t001:** Crystallite size, dislocation density, and lattice strain values of the Al-based metal matrix composites reinforced with different concentrations of HEA.

Composition	Crystallite Size (nm)	Dislocation Density (m^−2^)	Lattice Strain
Al6063–5 wt.% HEA	36 ± 0.45	1.59 × 10^15^	0.1663
LM25–5 wt.% HEA	37 ± 0.36	1.28 × 10^15^	0.1395
LM–10 wt.% HEA	35 ± 0.86	1.75 × 10^15^	0.2114

**Table 2 materials-15-00230-t002:** Microhardness and tensile properties of Al-based metal matrix composites with different concentrations of HEA reinforcement.

Condition	Microhardness (HV_0.20_)	Yield Strength (MPa)	Ultimate Tensile Strength (MPa)	Elongation (%)
As Cast LM25	55 ± 2	86 ± 2	130 ± 2	8 ± 0.4
Al6063–5 wt.% HEA	62 ± 3	160 ± 8	197 ± 1	4 ± 0.9
LM25–5 wt.% HEA	78 ± 6	150 ± 2	195 ± 1	3 ± 0.9
LM–10 wt.% HEA	92 ± 4	159 ± 3	210 ± 3	3 ± 0.3

## Data Availability

This data is a part of an ongoing study, and the data will be made available on reasonable request.

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
