# Peer review of "Evolution of Microstructure and Mechanical Properties of LM25–HEA Composite Processed through Stir Casting with a Bottom Pouring System"

_materials, 2021, doi:10.3390/ma15010230_

Round 1

Reviewer 1 Report

The article presents new knowledge and principles in the field of development and production of composite materials based on metal matrices

It is written clearly and legibly, the principles are explained understandably even for uninterested readers

On the whole, it is suitable for publishing

I see the only but quite serious shortcoming in the Materials and Methods chapter. I think it would be appropriate to give a schematic description of the experimental equipment for sample preparation. Please also describe in more detail the course of the experiment, the preparation of the melt and the composite material.

Author Response

The article presents new knowledge and principles in the field of development and production of composite materials based on metal matrices.

It is written clearly and legibly, the principles are explained understandably even for uninterested readers.

On the whole, it is suitable for publishing.

I see the only but quite a serious shortcoming in the Materials and Methods chapter. I think it would be appropriate to give a schematic description of the experimental equipment for sample preparation. Please also describe in more detail the course of the experiment, the preparation of the melt and the composite material.

We thank the reviewer for the criticism and positive comments. As suggested by the reviewer, a schematic description of the experimental equipment for sample preparation is introduced. In addition, as suggested the following information are included: (1) course of the experiment, (2) the preparation of the melt, and (3) composite materials. Hope thea included information in the revision is sufficient to warrant a publication in an esteemed MDPI journal like Materials.

Reviewer 2 Report

The authors have written on a topic titled ‘Evolution of Microstructure and Mechanical Properties of LM25-HEA Composite Processed Through Stir Casting with Bottom Pouring System’. Even though the research adds to existing knowledge on fabrication and characterization of aluminium based composites, aside some grammatical blunders that need to be fixed, the overall quality of the manuscript would be improved provided the authors respond accordingly to the following comments.

  1. Which type of stir casting technique was used for fabrication of the specimens? Was it manual or mechanical? The response should reflect in the abstract
  2. LM 25 is an aluminium alloy which is known for its improved strength and corrosion resistance. The authors should include more features and areas of application of this class of alloy in the introduction section.
  3. Several reinforcements such as carbides, nitrides and agro-based materials have been reportedly incorporated into the matrix of pure aluminium/aluminium alloys, however, the authors have chosen HEA(CoCrFeMnN) as reinforcement. What is the justification for the selection of CoCrFeMnN as reinforcement?
  4. Prior to casting, at what temperature was the reinforcement incorporated into the aluminium matrix?
  5. The last part of the methodology section should be changed to reported speech (line 106-109)
  6. From the optical micrographs, the authors emphasized on the distribution of silicon particles. However, silicon was not among the composition of the HEA reinforcement. The authors should attribute the presence of silicon to one of the major elements present I LM 25 alloy.
  7. Which figure is being referred to in line 199?
  8. Technical explanation should be provided on the XRD analysis. Was there the formation of new phases? If yes, mention them. Why is there a peak shift on the 2-theta axis even though it wasn’t very evident?
  9. From the fractography images, the authors should label the dimples and cleavage reported in the discussion

Author Response

Response to Comments of Second Reviewer

The authors have written on a topic titled ‘Evolution of Microstructure and Mechanical Properties of LM25-HEA Composite Processed through Stir Casting with Bottom Pouring System’. Even though the research adds to existing knowledge on fabrication and characterization of aluminium based composites, aside some grammatical blunders that need to be fixed, the overall quality of the manuscript would be improved provided the authors respond accordingly to the following comments.

Which type of stir casting technique was used for fabrication of the specimens? Was it manual or mechanical? The response should reflect in the abstract.

Bottom pouring type of stir casting technique with automated mechanical stirring was used for fabrication of the specimens. To make this clear a schematic representation is included in the revised version of the manuscript as Fig. 1.

LM 25 is an aluminium alloy which is known for its improved strength and corrosion resistance. The authors should include more features and areas of application of this class of alloy in the introduction section.

As suggested by the reviewer more information about LM25 aluminium alloy has been introduced in the revised version (Pl. refer to the Introduction part).

Several reinforcements such as carbides, nitrides and agro-based materials have been reportedly incorporated into the matrix of pure aluminium/aluminium alloys, however, the authors have chosen HEA (CoCrFeMnNi) as reinforcement. What is the justification for the selection of CoCrFeMnNi as reinforcement?

Aluminium/aluminium alloys reinforced with carbides, nitrides and agro-based materials have demonstrated marginal improvement in strength but showed significant chances of cracking during casting. On the other hand, choosing CoCrFeMnNi (metallic) as reinforcement helps in achieving required strength with no serious deterioration in ductility. CoCrFeMnNi reinforcement has higher thermal stability and can have better wettability and less diffusivity with the matrix.

Prior to casting, at what temperature was the reinforcement incorporated into the aluminium matrix?

The reinforcement was incorporated directly in to the aluminium matrix, when the melt was around 750-800 °C. No specific treatment was offered to the reinforcement.

The last part of the methodology section should be changed to reported speech (line 106-109).

Modified as suggested.

From the optical micrographs, the authors emphasized on the distribution of silicon particles. However, silicon was not among the composition of the HEA reinforcement. The authors should attribute the presence of silicon to one of the major elements present in LM 25 alloy.

As suggested, modifications were done on the revised version of the manuscript.

Which figure is being referred to in line 199?

It is Figure 7, which is referred to and included in the revised manuscript.

Technical explanation should be provided on the XRD analysis. Was there the formation of new phases? If yes, mention them. Why is there a peak shift on the 2-theta axis even though it wasn’t very evident?

We do not observe any formation of new phase from the XRD pattern. In addition, there is no peak shift on the 2-theta (peak position remains same) for the LM25 alloy composites. The peak shift is observed between LM25 composite and Al6063 composite, which is common due to change in lattice parameters even through all belong to fcc phase and Al-based systems.

From the fractography images, the authors should label the dimples and cleavage reported in the discussion.

As suggested by the reviewer dimples and cleavage in the fractography images were labelled.

Author Response

Response to Comments of Third Reviewer

Abstract:

Authors have summarized their work in abstract and summary of results included mechanical properties and fracture behavior.

A few more sentences should be added to abstract regarding characterization, phases identified, and the outcome of the research in terms of target applications of the composites.

The claim ‘Novel’ is unacceptable. The material cannot be claimed as Novel merely of the basis of composition unless there are some striking outcomes. Thousands of articles are available on Al/Al-alloy based composited reinforced with metallic and ceramic reinforcements.

As suggested by the reviewer, necessary changes were done in the abstract and the word ‘Novel’ was removed.

Introduction / Literature Review:

The Literature review is not appropriate. Authors do not communicate any recent or up-to-date information, neither they are able to highlight the research gaps properly which may prove their work relevant. Most of the text of Introduction is general and obvious, a thorough revision is required.

Only 24 references have been cited, authors need to enhance the literature review by cited more references. The article published in ‘materials’ may be included as some very informative experimental and review articles of metallic composites are available in materials.

Necessary modifications were done in the Introduction & Literature Review sections as recommended..

Materials and Methods:

Why 800 °C?

Response to reviewer comment: In the present work, LM 25 is the matrix material and can be heated in the range 680 – 710 °C. However, as the crucible used for casting was made of cast iron, we have increased the temperature and operated it at 800 °C.

Also, in order to have effective bonding between matrix and the HEA reinforcement, we have maintained a higher temperature of 800 °C.

Why 400 rpm?

We have gone through the literature and found that stirring was done in the range of 300 – 500 rpm. So we have chosen 400 rpm for our work based both on literature and on our own trial experiments.

Why stirring for 5-10 minutes?

In order to have effective bonding between matrix and the HEA reinforcement, we have considered stirring time of 5 – 10 minutes. If the stirring time is greater than 10 minutes, the melt may solidify and if it is less than 5 minutes, it may result in non-uniform reinforcement.

What about oxidation of the melt and temperature drop in this long duration?

As it is a fully mechanized, while stirring also, the furnace is continuously operative and temperature was being maintained. Hence, there is no possibility of drop in temperature. For avoiding oxidation, continuous low pressure Argon gas purging was done. These details are included in the Materials and Methods section.

At 800 °C, Aluminum oxidizes considerably? How it was prevented or controlled?

For avoiding oxidation, continuous low pressure Argon gas purging was done and the details are included in the Materials and Methods section.

Why bottom pouring? Is there any reason or just because of the availability of the set-up it was used?

Our set-up is designed for casting with bottom pouring facility as it helps in minimizing segregation of reinforcement phase at the bottom.

Which furnace was used?

We have used a stir casting furnace with bottom pouring facility having automated/controlled mechanical stirring facility.

Was there any treatment given to the particles give prior to dispersion?

We have gone through literature and found that no prior treatment was given to HEA particles and we too did not provide any treatment to the particles prior to dispersion.

On what basis the % of reinforcement was selected?

Based on the literature survey, most of the researchers used reinforcement in the range of 5 to 10 wt.% and obtained optimized results. This motivated us to choose reinforcement as 5% and 10 wt.%. Further, it is evident that for a less ductile system like Al-Si, adding more than 10 wt.% reinforcement phase can seriously affect the ductility of the final composite. On the other, being metallic reinforcement phase, less than 5 wt.% may not improve the strength to a recognized level.

Results and Discussion:

What do authors mean by HEA particles? How high entropy alloy is expected (or proven) to affect the properties? What are the advantages over low entropy alloy? How these metallic particles were classified as HEA? I mean what is the source of information?

HEA particles means alloy particles having high configurational entropy of the alloy system. This can be achieved by alloying 5 or more number of elements which are not having large difference in atomic size and other necessary requirements as suggested by earlier reports. As an alloy, high entropy alloy proved good with enhanced mechanical properties due to their sluggish diffusion kinetics as well as very high solid solution strengthening. These benefits couldn’t be obtained with low entropy alloy systems. Having more than 5 or more elements with equiatomic fractions, but still, showing a simple crystal structure like bcc or fcc is the evidence of formation of HEA system. In our particles, these are already proved. Necessary modifications were done in the revised manuscript.

Figure 1: How can Optical microstructures be used to identify the phases and Particles? It should be identified by SEM-EDS. So, it is advised that the optical microstructure should only mark the phases without naming them. While the names should be mentioned in the SEM images given in Figure 2.

Necessary modifications were done in the revised manuscript.

Accordingly, the text should be modified in results.

Text was modified in the Results and Discussion section in the revised manuscript.

The SEM images are beautiful and reveal a lot of information about the material.

Sincere thanks for the admiring complement.

Hardness is the primary mechanical property, it should be moved after XRD results. The hardness and microhardness have been reported at separate places, they should be reported and discussed together or one after the other. Normally the sequence of mechanical properties is Microhardness-Hardness-Tensile Properties-Ductility-Fractography.

As suggested by the reviewer, Hardness has been included after the XRD results. Actually it was microhardness that was measured. So to avoid confusion, the sub-section heading was modified as Microhardness studies.

Figure 10 should me marked and annotated. 10(a) may be replaced with t better image if available. Even though the magnification is not high, the images are blurred, the scanning may have been done better. If available the authors may substitute all these images with better ones.

We apologize for the inconvenience and agree with the Reviewer that SEM images were blurred. Poor scanning is due to the problem with SEM equipment. Unfortunately, we are unable to substitute these images with better ones.

The discussion of Figures is good and well presented.

Sincere thanks for the admiring complement.

Conclusions:

Remove ‘Novel’ from conclusion # 1

The word ‘Novel’ was removed from Conclusion# 1.

What was the reason that the particles were distributed homogenously? Give brief reason.

As the stirring was done at optimum speed of          400 rpm and for duration of 5-10 minutes, it resulted in homogeneous reinforcement of HEA particles in the matrix and this optimization was done in a scientific way and the details of which cannot be provided due to restrictions from funding agency.

Mention the hardness, strength also in conclusions.

As suggested by the Reviewer, hardness and strength were mentioned in Conclusions section.

What is next? How the research outcomes can be used for further research or applications? Whether the Al/LM25 CoCrFeMnNi composite substitute any prevailing materials or offer superior properties than some other material?

Yes, Al/LM25 CoCrFeMnNi composite substitute the prevailing materials and offer superior properties. Existing literature have considered ceramic particles as reinforcement and this results in vast variation of thermal expansion coefficient between ceramic particles and metal matrix, inferior wettability at the interface and the reaction at the interface reduce the properties of MMCs. The present work results in improving the mechanical properties without significant reduction in ductility.

Is there any research outcome which tells about the unique observation about processing?

The unique observation about processing is that for the hard and less ductile system like Al-Si, the mechanical strength can be increased through HEA particle reinforcement without losing the ductility significantly.

Round 2

Reviewer 3 Report

Comments to the authors:

The revisions in the article are thankfully received and noted. While most comments are addressed, the literature review still needs some improvement. To be specific add 5-10 most recent references to the materials, compositions, and processes used for similar composites.

The research gaps should be written in the summary of literature review, which should give reasons for the research design, selection of materials, methods, and compositions.

In the response, perhaps the authors didn’t get the following point right.

Which furnace was used?

‘We have used a stir casting furnace with bottom pouring facility having automated/controlled mechanical stirring facility.

Actually, I wanted to know whether a resistance furnace or induction furnace was used.

Was there any treatment given to the particles given prior to dispersion?

‘We have gone through literature and found that no prior treatment was given to HEA particles and we too did not provide any treatment to the particles prior to dispersion.’

There is always a possibility of moisture pick-up by the particles, this problem is more common in ceramics while comparatively less in metals. This may lead to hydrogen porosity in the composite deteriorating bonding and mechanical properties. To avoid moisture pick-up and also to reduce the temperature difference it is almost certain that the particles are heated before dispersion.

Author Response

(1) The revisions in the article are thankfully received and noted. While most comments are addressed, the literature review still needs some improvement. To be specific add 5-10 most recent references to the materials, compositions, and processes used for similar composites. The research gaps should be written in the summary of literature review, which should give reasons for the research design, selection of materials, methods, and compositions.

As suggested by the reviewer the literature part is further improved. 5-10 most recent references to the materials, compositions, and processes used for similar composites was included. The research gap is highlighted, which forms the basis for the present research design, selection of materials, methods, and compositions.

(2) In the response, perhaps the authors didn’t get the following point right.

Which furnace was used?

‘We have used a stir casting furnace with bottom pouring facility having automated/controlled mechanical stirring facility.

Actually, I wanted to know whether a resistance furnace or induction furnace was used.

An induction furnace was used for melting the composite and this information has now been included in the revised version of the manuscript.

(3) Was there any treatment given to the particles given prior to dispersion?

‘We have gone through literature and found that no prior treatment was given to HEA particles and we too did not provide any treatment to the particles prior to dispersion.’

There is always a possibility of moisture pick-up by the particles, this problem is more common in ceramics while comparatively less in metals. This may lead to hydrogen porosity in the composite deteriorating bonding and mechanical properties. To avoid moisture pick-up and also to reduce the temperature difference it is almost certain that the particles are heated before dispersion.

The HEA samples are relatively stable at room temperature and have sluggish diffusion behavior. Hence, we have not included any treatment of the powder material before the introduction into the melt. In any case, we would like to acknowledge that mostly the powder particles will have a 3-4 nm thick oxide layer along the surface and cannot be removed by any external treatment. Hence, we refrain from proceeding with any further external treatment, before the introduction of the powder into the melt.